# Research on the Control of Gastrointestinal Strongyles in Sheep by Using *Lotus corniculatus* or *Cichorium intybus* in Feed

**DOI:** 10.3390/pathogens12080986

**Published:** 2023-07-27

**Authors:** Călin-Alexandru Cireșan, Ileana Cocan, Ersilia Alexa, Liliana Cărpinișan, Cătălin Bogdan Sîrbu, Diana Obiștioiu, Beatrice Ana-Maria Jitea, Tiana Florea, Gheorghe Dărăbuș

**Affiliations:** 1Faculty of Veterinary Medicine, University of Life Sciences “King Michael I” from Timisoara, Calea Aradului 119, 300645 Timisoara, Romania; lilianacarpinisan@usvt.ro (L.C.); catalin.sirbu@usvt.ro (C.B.S.); beatrice.jitea@usvt.ro (B.A.-M.J.); tijana.florea@usvt.ro (T.F.); gheorghedarabus@usvt.ro (G.D.); 2Faculty of Food Engineering, University of Life Sciences “King Michael I” from Timisoara, Calea Aradului 119, 300645 Timisoara, Romania; ileanacocan@usvt.ro (I.C.); ersiliaalexa@usvt.ro (E.A.); 3Faculty of Agriculture, University of Life Sciences “King Michael I” from Timisoara, Calea Aradului 119, 300645 Timisoara, Romania; dianaobistioiu@usvt.ro

**Keywords:** *Lotus corniculatus*, *Cichorium intybus*, gastrointestinal parasitism, efficacy

## Abstract

The general practice of sheep farmers in gastrointestinal helminth control is based on the use of commercial drugs, making chemoresistance very common. Considering this, our study focused on the biological control of gastrointestinal parasitism using high-tannin plant hay. Three groups of 30 animals each were formed. The control group was additionally fed meadow hay, while the other two groups received chicory (group 2) and bird’s foot trefoil hay (group 3). The number of gastrointestinal strongyle eggs, shed through faeces (EPG), was surveyed for 28 days for all animals. The amounts of total tannins for meadow, chicory, and *Lotus corniculatus* hay supplements were 13.92 mg/g, 78.59 mg/g, and 94.43 mg/g, while their condensed tannin contents were 2.58 mg/g, 29.84 mg/g, and 15.94 mg/g, respectively. Compared to experimental day 0, there was an increase in EPG of 80.83% in the control group, a decrease of 24.72% in group 2, and a 20% decrease in group 3, by day 28. The *p*-value was <0.05 between group 1 and the other groups, showing significant differences between the control and experimental groups. The decrease in EPG rates in the experimental groups compared to the control group demonstrates an antiparasitic effect of *Lotus corniculatus* and chicory.

## 1. Introduction

Sheep farming is one of the most important branches of agriculture worldwide [1]. The profitability of sheep farming is closely linked to grazing, and thus contact of animals with parasitic elements during grazing cannot be avoided [2]. Although there are classical methods to reduce parasites on pasture (alternative grazing, rotation of pastures, etc.), these are not fully effective [3], and therefore, in the end, drugs are used for parasitological control [4].

Gastrointestinal helminths are the cause for some of the main parasitic diseases that lead to losses in sheep farming due to a decrease in their productive performance (milk, meat, wool), the increase in expenses generated by the use of treatments, and the mortality rate generated by massive infections [5]. These parasites cause nutritional deficiencies, and thus animals are prone to developing other concurrent diseases [6]. Among them, the most commonly encountered and most pathogenic nematode of ruminants is *Haemonchus contortus* [1,7], a parasite that exhibits chemoresistance to most anthelmintic products available on the market [6].

Chemoresistance is a concern in the management of gastrointestinal helminths [8,9,10]. This phenomenon has been recognized for more than 50 years [11]. Thus, in this regard, in 2010, in Brazil, Alfredo et al. confirmed helminth chemoresistance to several substances such as Levamisole, Moxidectin, Albendazole, Ivermectin, Nitroxinil, Disophenol, Trichlorfon, and Closantel following tests conducted on a flock of 5000 sheep [12]. Similarly, chemoresistance to ivermectin has been confirmed in several areas of the world, such as Europe [13], Cuba [8], or Argentina [9].

The effect of medicinal plants in antiparasitic treatments has been known since ancient times [14]. Therefore, as an alternative to helminth chemoresistance, over time, the antiparasitic effects of several plants with tannin content (*Lotus corniculatus, Cichorium intybus, Onobrychis*, etc.) have been tested [15,16,17,18,19]. The antiparasitic effect of these plants has been tested over the years both by direct administration of the plant as such [15] or by using plant extracts containing antiparasitic substances [17,18]. 

*Lotus corniculatus* (bird’s foot trefoil) is an herbaceous perennial plant belonging to the legume family. It can grow wild in meadows, but it can also be used as a fodder plant and cultivated. Bird’s foot trefoil is widespread throughout the world and adapts to any type of soil or climate. The main benefit of this plant is that it does not cause tympanism in ruminants, regardless of how it is administered to the animals [20]. *Cichorium intybus*, known simply as chicory, is a widespread plant throughout Europe and Asia and has been recognized for its anthelmintic effects since ancient times. It belongs to the *Asteraceae* family, *Chicorium* genus, and is a perennial plant that can reach up to 1 meter in height and is very resistant to temperature changes both during the vegetative and reproductive periods. It is a non-toxic plant, and although it has long been known in traditional medicine for its wide therapeutic usefulness, many of its components have not yet been sufficiently studied [16].

Condensed tannins are plant secondary compounds that bind strongly to proteins following ruminant mastication, thereby reducing their degradation in the rumen at pH 6–7. The protein complex dissociating at pH lower than 3.5 in the abomasum increases the absorption of essential amino acids in the small intestine of sheep that were fed bird’s foot trefoil. These tannins have a beneficial effect as they reduce bloating and greenhouse gas production following their binding to proteins [21,22,23]. Due to the demonstrated beneficial effect of tannin on digestibility and its inhibitory effect on helminth development, most studies have focused on diets including tannin-rich plants. In temperate agriculture, there is a need to develop technologies that incorporate tannin-rich plants into animal diets, while in tropical and humid areas, the leaves of tannin-rich trees and shrubs should be used [24].

The aim of our study was to determine the levels of total and condensed tannins in the hay fed to sheep (wildflower meadow hay, chicory hay, and bird’s foot trefoil hay) and their usefulness in the parasitological control of diseases caused by gastrointestinal strongyles. In order to observe the effect that these plants have on parasites, we made use of hay produced directly from them to mimic real farm conditions as closely as possible.

## 2. Materials and Methods

### 2.1. Ethic Statements

The study was carried out on animals (90 sheep) in compliance with the legislation in force in the country regarding experimentation on animals and with the approval of the university’s bioethics committee (document number 160/2022).

### 2.2. Determining the Total and Condensed Tannins

For the determination of total and condensed tannins, samples were taken from each category of hay intended to be fed to the sheep in the experiment (wildflower meadow hay, chicory hay, and bird’s foot trefoil hay). Three samples were taken for each category of hay (9 hay samples in total) for the determination of their tannin content. Tannin determinations were performed in the laboratories of the Food Engineering Faculty in Timisoara.

#### 2.2.1. Extraction of Total Tannins

Tannins were extracted using the cold percolation method. Thus, 1 g of each sample of hay was suspended in 6 mL of 70% acetone solution containing 0.01% ascorbic acid stored in airtight containers in a water bath for 10 h at 50 °C. Samples were subsequently agitated using linear movements (DLAB SK-L330-Pro, Beijing, China) for 3 h. Both steps were repeated twice, and in the end, the samples were maintained in a water bath for 4 h at 50 °C. The processed samples were filtered using Whatman N°1 filter paper and dried in a room temperature air-flow room, followed by a 1 h drying period in a thermostat at 50 °C.

#### 2.2.2. Determining Total Tannins

The method of Petchidurai (2019), with minor modifications, was used for the determination of total tannins [25]. Thus, 0.1 g of extract were dissolved in 10 mL of Folin Ciocalteu solution (Sigma-Aldrich Chemie GmbH, Munich, Germany) diluted with water 1:20 (*v/v*). The obtained mixture was incubated for 3 min at room temperature, followed by the addition of 2 mL of Na_2_CO_3_ 35% aqueous solution. The final solution was re-incubated for 30 min before reading the absorbance at 725 nm using a Specord 205 UV-VIS spectrophotometer from Analytik Jena Inc. (Jena, Germany). The total tannin content was expressed on the basis of the tannic acid standard curve, which was indicated in mg/gram in the range 20–200 mg/g. 

#### 2.2.3. Extraction of Condensed Tannins

After centrifugation, the supernatant containing the condensed tannins was stored in vials in a refrigerator for further studies. The total condensed tannin content was expressed on the basis of a standard gallic acid (GAE) or tannic acid (GTA) curve as milligrams × per gram of GAE or GTA, respectively.

For the extraction of condensed tannins, hay samples were ground with a Grindomix GM 2000 laboratory mill (Retsch GmbH, Haan, Germany), followed by the collection of 1 g of dried and ground material. Six mL of 70% acetone containing 0.01% ascorbic acid and 3 mL of acetone: water: diethyl ether mixture (4.7 : 2.0 : 3.3) were added. The resulting mixture was centrifuged at 15,000 rpm for 30 min to recover the condensed tannin and remove residual pigments. Centrifugation resulted in the separation of the solution into two phases (the aqueous phase and the acetone containing phase). The supernatant was taken for the determination of the condensed tannins.

#### 2.2.4. Determination of Condensed Tannins

The quantification of condensed tannin was carried out using the Makkar (2003) method with Gallic acid and tannic acid as standards [26]. Absorbance at 725 nm was read using a spectrophotometer (Eleco, SL 171 Mini Spec, Kolkata, India) after incubation of the reaction mixture at room temperature for 40 min using a Specord 205 UV-VIS spectrophotometer, Analytik Jena Inc. (Jena, Germany). 

Condensed tannins were quantified according to the method of Makkar (2003) using tannic acid as a standard. Thus, the final solutions prepared according to the method for the determination of total tannins were again incubated for 40 min at room temperature, after which the absorbance rate was determined at 725 nm using a Specord 205 UV-VIS spectrophotometer, Analytik Jena Inc. (Jena, Germany). The different tannin contents were expressed in mg tannic acid equivalents/g using the tannic acid curve in the concentration range 20–200 mg/g.

#### 2.2.5. Statistical Data Analysis

A one-way ANOVA followed by a two sample t-test with equal variance were used to evaluate statistical differences for total and condensed tannins. Three determinations were performed for each analysis in the case of each sample, and the results were expressed as the mean of the three determinations ± standard deviation (SD). Microsoft Excel 365 (version 2208, Redmond, WA, USA) was used for the statistical processing of the data.

### 2.3. Control of Gastrointestinal Strongyles through Administration of Cichorium intybus and Lotus corniculatus Hay

#### 2.3.1. Experimental Groups

The study was conducted on animals from a private farm located in Tormac commune, Timis county, western Romania, over a 28-day period and was carried out on three groups of 30 *Turcana* breed sheep, about 1 year of age. All 90 sheep were females, with body weights ranging from 40 to 50 kg, evenly distributed in the three groups. The sheep were weighed at the beginning of the study and at the end to see whether there were any differences between the average weights of the sheep groups. Weighing was carried out individually for each animal using an electronic scale (Cima, Control in Motion, Italy). The ewes came from the same farm and were acclimatized to the conditions found on the farm. They were vaccinated 6 months prior to the study against anthrax, according to the mandatory strategic program provided by the national legislation (the pharmaceutical product used for this vaccination was Carboromvac, produced by Romvac, Romania), and against Clostridiosis (the commercial product used for this vaccination was Coglavax, produced by Ceva, France). The last internal deworming of these animals was also carried out about 6 months before the study by oral administration of albendazole (commercial product Vermitan, produced by Ceva, France), and these animals received Diazinon (commercial product Diazinol, produced by Pasteur, Romania) for their external deworming about 3 months prior to the study. The animals were reared in a semi-open system. Throughout the experimental period, the animals grazed together on a pasture with the spontaneous flora characteristic of the lowland area. During the night, the three groups were placed in a stable system and were given additional feed, which they consumed in its entirety before returning to pasture the following day. The supplementary feed, depending on the flock, was:One meal of meadow hay of approximately 20 kg/group/day for the control group (group 1);One meal of chicory hay of approximately 20 kg/group/day for group 2;One meal of bird’s foot trefoil hay of approximately 20 kg/group/day for group 3.

All of this hay was produced on the farm in their own fields. As the soil where the animals grazed is poor in selenium, the animals were given salt blocks with added selenium ad libitum both before the experimental period and during the experiment. The source of water for the animals was represented by a well drilled at a depth of 15 m on the farm.

#### 2.3.2. Sample Collection and Examination

Faeces were collected directly from the rectum of each sheep from all three groups on days 0, 7, 14, 21, and 28 of the experiment. A total of 450 faecal samples were collected throughout the study. The samples were individualized and refrigerated until examination. The maximum length of time that these samples were refrigerated for was 48 h, during which time the samples from the three groups of animals were examined in parallel to avoid influencing the results due to different times of examination. They were examined in the laboratories of the Parasitology Clinic of the Faculty of Veterinary Medicine in Timisoara. The testing was conducted using the McMaster method, while the anthelmintic efficacy rate was determined using the FECRT [27]: $$E\%=\frac{E.P.G.\;\mathrm{before}\;\mathrm{treatment}\;\left(\mathrm{day}\;0\right)-E.P.G.\;\mathrm{day}\;x}{E.P.G.\;\mathrm{on}\;\mathrm{day}\;0}\times 100$$

In order to perform the McMaster method, two grams of faeces were taken from each sample collected from sheep and soaked in 10 mL of saturated NaCl solution. The resulting mixture was then transferred into a measuring cylinder, where a NaCl saturated solution was added up to 30 mL. After making the final mixture, the contents were poured into a plastic beaker and shaken well in order to homogenise the solution as adequately as possible. Finally, the 2 chambers of the McMaster slide were filled as evenly as possible, without air bubbles, with the help of a pipette and examined microscopically within the grids of each chamber using the 10× objective (Microscop Bresser Researcher Bino 40–1000×, Germany). The analytical sensitivity considered was 20 eggs/oocyst per gram. The number of eggs per gram of faeces (EPG) was calculated using the equation:$$E.P.G=\frac{n\;\times 100}{2}.$$
where “n” equals the number of eggs counted in both chambers of the McMaster slide. Eggs that did not fall between the McMaster chamber lines were not counted [28]. 

Throughout the experiment, blood samples were collected from 5 animals pertaining to each experimental group on days 0, 14, and 28 in tubes with anticoagulant (EDTA) to determine the complete blood count and its evolution during the 28 experimental days. Immediately after collection, the blood was transported in a cooler at 2–8 °C and examined within a few hours in a private laboratory. These blood samples were taken to observe if there were any significant changes in the sheep’s blood count.

#### 2.3.3. Statistical Data Analysis

All statistical data related to the evolution of the EPG of the 3 groups of animals as well as the differences between the 3 groups were performed using statistical functions implemented in Microsoft Excel 365 (version 2208, Redmond, WA, USA). To assess the statistical differences between the groups of animals, a t-test with different variances was used both between the control group and each experimental group and between the 2 experimental groups.

## 3. Results

### 3.1. Determination of Total and Condensed Tannins

The results of the total and condensed tannin determinations for all the types of hay are shown in Figure 1. Noticeably higher amounts of total and condensed tannins can be found in chicory and bird’s foot trefoil hay compared to meadow hay. Also, despite the fact that bird’s foot trefoil contains higher amounts of total tannins than chicory, the amount of condensed tannins is higher in chicory hay.

### 3.2. Control of Gastrointestinal Strongyles via Administration of Cichorium intybus and Lotus corniculatus Hay

The evolution of the sheep’s body weight can be observed in Table 1. 

The evolution of the total weight of each group of animals did not follow the same trend, i.e., the weight of the control group was slightly declining compared to the weight of the other two groups of sheep, where it was slightly rising. 

The evolution of eggs/gram of faeces (EPG) following the use of meadow, chicory, and bird’s foot trefoil hay in terms of parasitic control of gastrointestinal strongyles is shown in Table 2.

The evolution of the parasitic load can be observed in Figure 2 (the average EPG for each group).

The percentage evolution of parasite egg shedding determined by using the FECRT count method is presented in Figure 3. 

A progressive increase in gastrointestinal strongyle egg shedding can be observed in the group receiving additional meadow hay, reaching 80.83% on day 28. At the same time, in the groups that received additional chicory hay or bird’s foot trefoil hay, although the grazing took place under identical conditions as for the control group, a decrease in gastrointestinal strongyle egg shedding rates could be seen.

Chicory hay reduced the parasite load of the experimental group by 21.57% after 7 days, 37.79% after 14 days, 25.19% after 21 days, and 24.75% after 28 days of administration.

The bird’s foot trefoil hay reduced the parasite load of the test groups by 19.16% after 7 days, 31.18% after 14 days, 23.49% after 21 days, and 20% after 28 days.

According to the t-test, the difference between group 1 and the other groups was not significant on day 0 (*p* > 0.6 between group 1 and group 2 and *p* > 0.7 between group 1 and group 3). Right until the end of the experiment, the *p*-value was <0.05 between group 1 and the other groups, showing significant differences between the control and experimental groups. This aspect is absent when observing the *p*-value between experimental groups 2 and 3. Throughout the entire experiment period, the *p*-value ranged from 0.17–0.43 for these groups, demonstrating that there were no significant differences among the groups fed with chicory hay and bird’s foot trefoil hay. 

The blood parameters of the sheep in the three groups of animals can be seen in Table 3. 

The animals in the experimental groups showed a higher content of red blood cells and haemoglobin towards the end of the experiment, but the differences are relatively small, not significant, as when comparing the groups statistically, the *p*-value is always >0.05. Between the two experimental groups, the differences are much smaller.

## 4. Discussions

There is various data in specialty literature reporting different amounts of condensed tannins found in plants, depending on the extraction method or the developmental stage of the plant. Thus, some authors report concentrations of 3 mg/g for chicory and 15.2 mg/g for bird’s foot trefoil [15], while others report concentrations of 13.1 mg/g and 28.2 mg/g [30]. The present study, however, reveals contradictory differences regarding the amounts of condensed tannins for the studied plants, with higher concentrations for chicory (29.84 mg/g) compared to bird’s foot trefoil (15.94 mg/g). It is a well-known fact that condensed tannins do not depend only on the type of plant considered but also on the time, place, and season of harvest, etc. Thus, differences among the same species of fodder plants may occur.

This study used tannin-containing plants, which were administered as hay, in an attempt to biologically control gastrointestinal strongyles and benefit farmers without having to use plant extracts. The diet was not entirely made up of chicory or bird’s foot trefoil in the case of the experimental groups because the findings from a previous study [31] showed the reluctance of animals to consume the necessary amount of feed in order to meet their nutritional requirements due to the bitter taste of tannin-containing plants, leading to low rates of weight gain or even weight loss. Throughout the entire experimental period, the animals from all three groups failed to present any appetite alterations, despite any differences in the ratio, and no clinical diseases that could interfere with the final results of the experiment were diagnosed during the experimental period.

Juhnke et al. (2012) demonstrated that lambs, when parasitized with *H. contortus*, prefer to eat tannin-containing plants [32]. However, excessive intake of tannin-rich plants was associated with a reduction in feed intake. Similarly, other experiments have shown that administering condensed tannins to ruminants, while reducing their parasitism levels, did not improve the productive performance of the animals [33], e.g., feeding *Lotus corniculatus* can reduce the fat concentration in sheep milk [34]. Other studies that have used tannin-containing plants have shown an increase in milk yield in ruminants but also a decrease in milk quality due to the bitter taste that milk acquires as a result of excessive tannins [19]. The economic benefits, however, might be more pronounced with direct grazing of the plants. Thus, it has been shown that sheep grazing on tannin-rich pastures showed low levels of parasitism and improved their productive performance compared to those grazing on simple pastures without tannin-rich plants [35].

In our study, the animals were exposed to pasture-linked parasites because during the daytime, the animals were out grazing, thus resulting in a slight increase in the EPG at 28 days in the experimental groups and a higher increase in group 1, which received meadow hay. Although bird’s foot trefoil or chicory did not account for 100% of the feed given to the animals in groups 2 and 3, their parasite load level throughout the experiment remained lower than the level of the control group. Depending on the different parasite loads observed in the three groups of animals, the weight of the animals also varied, so in the animals from group 1, where the parasite load increased, the body weight dropped, while in the other two experimental groups, the opposite was observed. This increase in body weight in animals from groups 2 and 3 may be associated with a lower parasite load.

The increased RBC content in the animals from the experimental groups can be attributed to a weaker parasite infection compared to the parasite load in the control group. Regarding these differences between the blood counts of the control animals compared to the experimental animals, other similar studies have demonstrated greater variations [36].

Regarding the antiparasitic effectiveness, a similar study found that tannin-containing plants cause significant reductions in total daily faecal *H. contortus* egg production (chicory: 89%; bird’s foot trefoil: 63%; sainfoin: 63%), probably due to the reduction in adult *H. contortus* (chicory: 15%; bird’s foot trefoil: 49%; and sainfoin: 35%). When the feeding of tannin-containing plants was stopped, no significant increase in the number of faecal eggs was found, suggesting that the reduction in parasite egg production is sustainable [15]. Another study reveals that by grazing bird’s foot trefoil, lower parasite populations were found in the abomasum and small intestine of lambs. It is not clear from the experiment why tannins have a stronger effect on abomasum parasites, but changes in the normal physiology or immunological functionality of the abomasum could be involved [37]. It also appears that lambs grazing bird’s foot trefoil have a lower parasite load in the abomasum and small intestine compared to lambs grazing white clover (*Trifolium repens*) [35] and also have a higher average daily weight gain [38].

Other studies have shown that reduction of parasite load can also be achieved by administering plant extracts with high tannin content, resulting in up to 64% reduction in *H. contortus* egg shedding/gram of faeces in goats [39], up to 96.5% reduction using bird’s foot trefoil-based extracts [17], and recently, in 2022, up to 30.3% efficacy against *H. contortus* was demonstrated, also using tannin-based plant extracts [18], the latter value being close to that in our experiment. This decrease in the number of eggs/gram of faeces can be attributed to the larvicidal effect exerted by the plant extract based on condensed tannins on larvae 3 of *H. contortus* [6] as well as to the negative effect of these tannins on trichostrongylus females, represented by low fecundity after exposure to condensed tannins [15].

## 5. Conclusions

Bird’s foot trefoil and chicory hay contain higher amounts of total and condensed tannins compared to wildflower meadow hay.

The reduction of EPG in the experimental groups compared to the control group demonstrates an antiparasitic effect of bird’s foot trefoil and chicory.

Chicory and bird’s foot trefoil hay can be used as feed supplements for grazing sheep for the control of gastrointestinal strongyle populations. This protocol of combining direct grazing with wild flora and tannin-containing plant supplements could be used for both traditional and organic farming.

## Figures and Tables

**Figure 1 pathogens-12-00986-f001:**
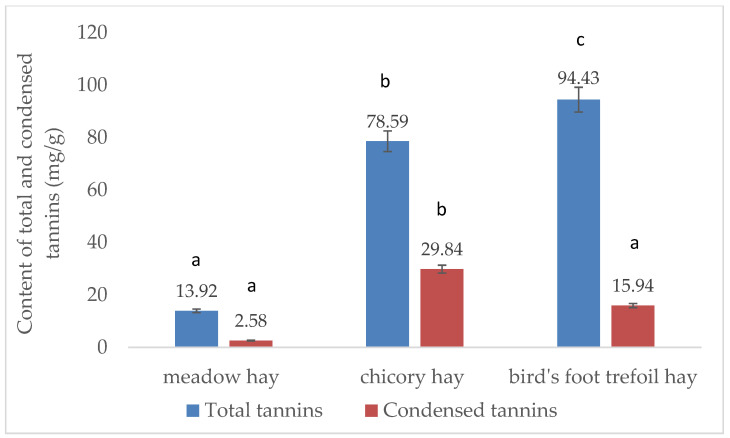
Content of total and condensed tannins in the studied hay samples (n = 3). The results are expressed as the average value of three determinations, ±the standard deviation (SD) indicated by the error bars. According to the *t*-test, the different letters (a–c) represent the significant differences (*p* < 0.05) between the values presented in the columns.

**Figure 2 pathogens-12-00986-f002:**
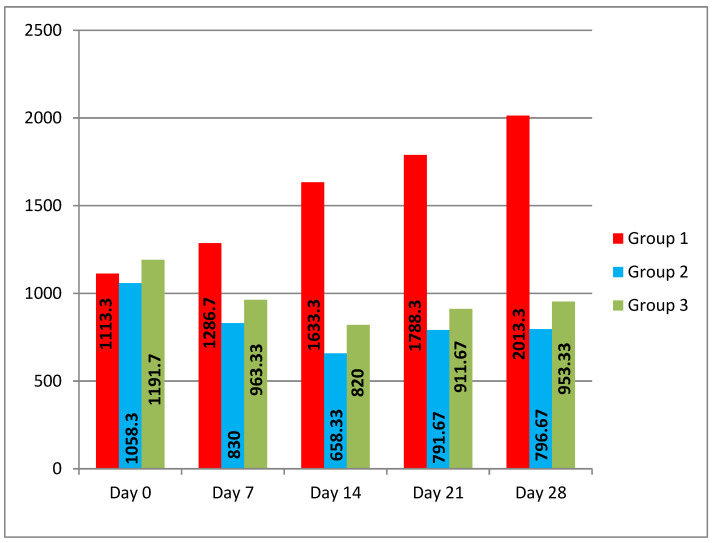
Evolution of the parasitic load throughout the experiment (Average EPG/group) for all three groups (n = 30).

**Figure 3 pathogens-12-00986-f003:**
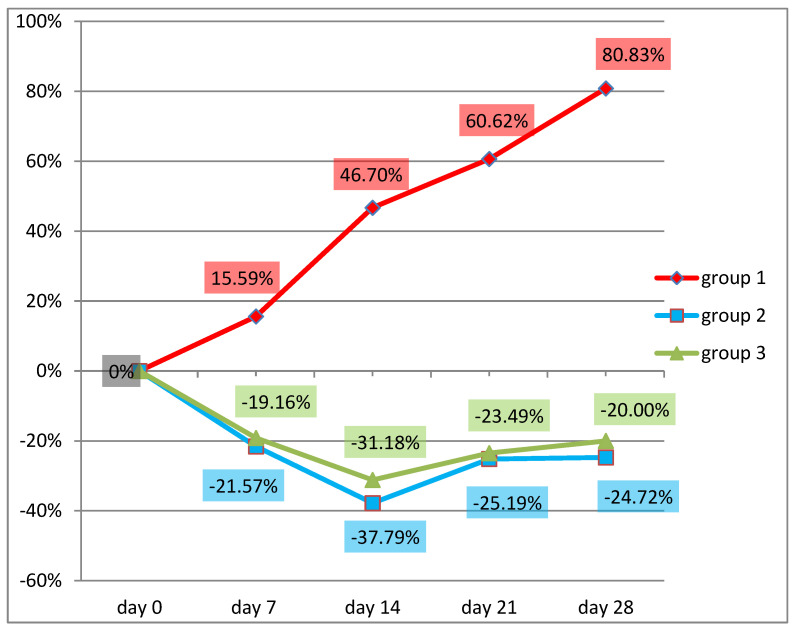
The percentage evolution of the parasitic load of all three groups compared to day 0 (n = 30).

**Table 1 pathogens-12-00986-t001:** Body weight evolution in sheep from all three study groups.

	Group 1 (Control)—Meadow Hay		Group 2—Chicory Hay		Group 3—Bird’s Foot Trefoil Hay	
	day 0	day 28	day 0	day 28	day 0	day 28
Total Kg/group	1372.5	1359.5	1339	1383.4	1357.9	1406.3
Minimum individual weight	40.2	40.2	39	40.2	40.4	41.5
Maximum individual weight	49.6	50	49.6	52.2	49.1	49.5
Mean	45.75	45.31	45.26	46.87	44.63	46.11
Standard deviation	3.04	2.84	3.41	3.52	2.81	2.32
Mean standard error	±0.55	±0.51	±0.62	±0.64	±0.51	±0.42

For each daily determination, 30 samples/group were examined, namely 1 sample from each sheep (n = 30).

**Table 2 pathogens-12-00986-t002:** Evolution of the EPG for all the groups.

	**Group 1 (Control)** **—** **Meadow Hay**				
	day 0	day 7	day 14	day 21	day 28
Total EPG/group	33,400	38,600	49,000	53,650	60,400
Minimum	0	0	0	0	300
Maximum	2550	2550	3000	3650	3950
EPG average/animal	1113.3 ^a^	1286.7 ^a^	1633.3 ^a^	1788.3 ^a^	2013.3 ^a^
Standard deviation	632.31	627.93	811.7	871.06	945.13
Mean standard error	±115.44	±114.64	±148.19	±159.03	±172.56
	**Group 2** **—** **Chicory Hay**				
	day 0	day 7	day 14	day 21	day 28
Total EPG/group	31,750	24,900	19,750	23,750	23,900
Minimum	0	0	0	0	0
Maximum	3250	1900	1650	2000	1750
EPG average/animal	1058.3 ^a^	830 ^b^	658.33 ^b^	791.67 ^b^	796.67 ^b^
Standard deviation	695.81	513.54	370.95	506.52	485.29
Mean standard error	±127.04	±93.759	±67.725	±92.478	±88.601
	**Group 3** **—** **Bird’s Foot Trefoil Hay**				
	day 0	day 7	day 14	day 21	day 28
Total EPG/group	35,750	28,900	24,600	27,350	28,600
Minimum	0	0	0	0	0
Maximum	2750	2450	1850	1750	1850
EPG average/animal	1191.7 ^a^	963.33 ^b^	820 ^b^	911.67 ^b^	953.33 ^b^
Standard deviation	619.8	595.95	517.55	533.48	511.41
Mean standard error	±113.16	±108.81	±94.492	±97.4	±93.37

According to the *t*-test, the different letters (a–b) represent the significant differences (*p* < 0.05) between the groups for all 28 days. For each daily determination, 30 samples/group were examined, namely 1 sample from each sheep (n = 30).

**Table 3 pathogens-12-00986-t003:** Blood counts of sheep from the 3 groups of animals over the course of the experiment.

	**Group 1 (Control)** **—** **Meadow Hay**			
	day 0	day 14	day 28	**Ref.**
n	5	5	5	
RBC (×10^6^/mL)	9.81 ± 1.69	8.17 ± 1.56	7.56 ± 1.37	9–15
HGB (g/dL)	10.7 ± 1.81	9.4 ± 1.81	9.3 ± 2.05	9–15
PCV (%)	27.2 ± 3.45	25.2 ± 2.35	27.5 ± 4.63	27–45
WBC (×10^3^/mL)	8.98 ± 2.38	8.85 ± 2.18	13.39 ± 4.77	4–12
	**Group 2** **—** **Chicory Hay**			
	day 0	day 14	day 28	**Ref.**
RBC (×10^6^/mL)	9.8 ± 3.32	10.08 ± 1.88	11.36 ± 3.03	9–15
HGB (g/dL)	12.6 ± 2.98	11.4 ± 2.21	11.8 ± 2.73	9–15
PCV (%)	34.5 ± 6.45	30.6 ± 9.31	30.6 ± 6.47	27–45
WBC (×10^3^/mL)	10.42 ± 3.21	12.1 ± 3.05	12.19 ± 2.86	4–12
	**Group 3** **—** **Bird’s Foot Trefoil Hay**			
	day 0	day 14	day 28	**Ref.**
RBC (×10^6^/mL)	10.12 ± 1.70	9.53 ± 1.35	10.69 ± 3.32	9–15
HGB (g/dL)	11.6 ± 1.77	11.1 ± 1.07	11.2 ± 3.35	9–15
PCV (%)	30 ± 2.19	28.7 ± 3.61	30.5 ± 4.38	27–45
WBC (×10^3^/mL)	10.96 ± 1.34	10.72 ± 3.31	10.87 ± 4.42	4–12

For each daily determination, 5 samples/group were examined (n = 5); the value in the table represents the average of the 5 samples; followed by the standard deviation. RBC: red blood cells; HGB: haemoglobin; PCV: packed cell volume; WBC: white blood cells; Ref: reference values according to Byers and Kramer (2010) [29].

## Data Availability

The report of the analyses performed for the samples in the paper can be found at the Interdisciplinary Research Platform (PCI) belonging to the University of Life Sciences “King Michael I” from Timisoara.

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
