# Peer review of "Research on the Control of Gastrointestinal Strongyles in Sheep by Using Lotus corniculatus or Cichorium intybus in Feed"

_pathogens, 2023, doi:10.3390/pathogens12080986_

Round 1
Reviewer 1 Report
The manuscript entitled ‘Research on the control of gastrointestinal strongyles in sheep by using Lotus corniculatus or Cichorium intybus in feed’ is well written and provides new insight for the biological control of gastrointestinal strongyles with high tannin plant hay.
There are some minor concerns that the author should address:
In the abstract section the it should be specified which of the two groups in addition to the control group was fed with bird’s foot trefoil and which with chicory hay. Please specific it in brackets to make for ease reading.
Please, write H. contortus elsewhere in the manuscript after the first mention in full in the introduction.
In the Material and Methods section and in particular in the paragraph 2.3.2 Sample collection and examination the are some point to review. At the end of the first sentence the period is missed. Furtermore, what is the analytical sensitivity of this technique? I presume it was 15 eggs/oocyst per gram but in my opinion it better to specify it. In addition, it is not very clear the method of reading used even if from the formula I can deduce that the reading was carried out within the grids of each chamber. Please, specify better the method used.
In the part regarding the blood examination there are two sentences a bit redundant: ‘Throughout the experiment, blood samples were collected in tubes with anticoagulant (EDTA) from all three groups of animals to determine the complete blood count and its evolution during the 28 days’ and ‘Theese blood samples were taken to observe if there are any significant changes in the sheep’s blood count’. Please combine these two sentence in one.
In the section Results I suggest to specify in the tables what is minimum and maximum, for example minimum individual weight, maximum individual weight etc.
Reviewer 2 Report
This is a very interesting study, well designed and conducted, the objetives are clear and results are well presented and accurately analysed. It is a work that contributes to broaden the knowledge on sustainable parasite control in livestock. I only have few minor comments on this manuscript:
No numbered lines were included in the manuscript which makes the review process difficult to perform and to indicate location of possible suggestions: Page 5 Typo Error “Theese”
Abstract: I suggest authors to mention if differences between groups were or were not statistically significant
Was blood sample taken from the same 5 animals of each group along the course of the study?
I wonder why authors did not consider to carry out parasite identification whether by performing PCR or Coproculture of strongyle eggs and L3 identification to specie level. These might have been a relevant data which may have illustrated the specific efficacy of Lotus corniculatus or Cichorium intybus on the different gastrointestinal strongyles.
Since this is a study on helminth infection and blood samples were taken to analyse haematological parameters, I also wonder why authors did not evaluate the different white cell types (i.e. eosinophil cell population) or plasma protein level along the course of the study.
